# Maternal educational level and preterm birth: Exploring inequalities in a hospital-based cohort study

**Laura Granés**[1][☯]*, **Isabel Torà-Rocamora**[1,2][‡], **Montse Palacio**[3][‡], **Laura De la Torre**[1][‡], **Anna Llupià**[1,2,4][☯]

**1** Preventive Medicine and Epidemiology Department, Hospital Clínic de Barcelona, Barcelona, Spain, **2** School of Medicine, Universitat de Barcelona, Barcelona, Spain, **3** Maternal-fetal Department, Institute of Gynecology Obstetrics and Neonatology, Hospital Clínic de Barcelona, Barcelona, Spain, **4** ISGlobal-Global Health Institute, Barcelona, Spain

☯ These authors contributed equally to this work.
‡ IT-R, MP and LDT also contributed equally to this work
* lauragranes@gmail.com

**Data Availability Statement:** Data cannot be shared publicly because of ethical considerations. Data are available from the Hospital Clínic Data Base upon request addressed to the Clinical Trials

## Abstract

Preterm birth has been related to inequalities in maternal educational level, but the causal mechanism is not entirely known. Some factors associated with preterm birth and low educational level such as chronic medical conditions, pregnancy complications and related-health behaviours could have a mediation role in the pathway. This study aimed to evaluate the association between maternal educational level and preterm birth, analysing the mediation role of these factors. We performed a retrospective cohort study based on hospital electronic records of 10467 deliveries that took place in the Hospital Clínic of Barcelona between 2011 and 2017. Poisson regression was used to obtain crude and adjusted relative risk of preterm birth in women with different educational level and the percentage of change in relative risk was calculated when mediation variables were included in the model. Women with a lower educational level had a higher risk of preterm birth (RR 1.57, 95% CI 1.21, 2.03). The loss of association after the inclusion of body mass index in the model suggests an important mediation role of maternal overweight. Other variables such as smoking, drug use, preeclampsia and genitourinary infections also appear to play a role in the observed inequality between women with different levels of education. Efforts to promote health literacy and to improve preventive interventions, before and during pregnancy, could decrease preterm birth rates and perinatal health inequalities

## Introduction

Preterm birth is defined as all birth occurring before 37 completed gestational week and it is the leading cause of neonatal mortality worldwide [1] and of infant mortality and loss of quality-adjusted life years in high-middle income countries [2]. The etiological mechanisms of preterm birth are complex and some causal pathways are still unknown. Smoking [3], alcohol consumption [4] and drug use [5] during pregnancy have been associated with an increased

and Research division of the Preventive Medicine
and Epidemiology Department, Hospital Clínic de
Barcelona (email address: cvac@clinic.cat)

**Funding:** The authors received no specific funding
for this work.

**Competing interests:** The authors have declared
that no competing interests exist.

risk of being born preterm. Chronic diseases such as hypertension [6], diabetes [7] and obesity [8] increase the probability of preterm delivery, especially when more than one factor is present [6]. The onset of pregnancy complications such as pregnancy-induced hypertension [9], preeclampsia [10], gestational diabetes [11], anemia [12], urogenital tract infections [13] and intrauterine infections [14] also raise preterm birth risk. Moreover, an adequate prenatal care has shown possible benefits in reducing preterm birth [15].

Heterogeneity in preterm birth rates have been found according to socioeconomic status in several studies and different countries [16–18]. In Barcelona, where preterm birth rate is around 6.5% overall, inequal preterm birth rates among neighbourhoods have been observed [19]. Some studies have found that women with a low educational level had worse pregnancy outcomes [20–22] compared with high educated women. Evidence suggests that educational level is more clearly related to inequalities in preterm birth than paternal educational level, occupation and household income [21].

Many variables related to higher risk of preterm birth such as substance abuse [23], overweight and obesity [24, 25], chronic health conditions [26], pregnancy complications [27, 28] and poorer use of prenatal health care services [29] have been associated with a low socioeconomic status. These variables, related with the exposure and outcome, could play a mediation role in the pathway of preterm birth inequalities. This study aimed to evaluate the association between maternal educational level and preterm birth analysing the role of those variables that could have a mediation role in the causal pathway.

## Methods

### Data sources and study population

We carried out a retrospective cohort study using the maternity hospital database of the Hospital Clínic of Barcelona. We included women who were followed up during their pregnancy and gave birth at our hospital between January 2011 and December 2017. The Hospital Clínic of Barcelona is a public high complexity hospital that provides care to two different populations. On the one hand, all women living in four districts of Barcelona city are assigned to the Hospital Clínic for their pregnancy follow-up, regardless of their obstetric risk. On the other hand, Hospital Clínic assists women referred due to high-risk pregnancies from other hospitals in semi-urban or rural areas. In order to minimise the selection bias inherent in hospital-based studies, for this study we only included women of the first group, who lived in Barcelona and were a priori assigned to our hospital. We excluded pregnancies terminated before 22 completed gestational weeks, multiple pregnancies and legal terminations of pregnancy after 22 gestational weeks but we included stillbirths as they share causal mechanisms with preterm birth [30].

This study was approved by the Clinical Research Ethical Committee of the Hospital Clínic (HCB/2019/0145).

### Exposure and outcome definitions

Maternal educational level was collected in the first pregnancy visit and considered in 3 categories: low (without studies and primary studies), medium (secondary level) and high (university or more). Preterm birth was defined as any birth after 22 and before 37 completed gestational weeks. Gestational age, collected from hospital database, was calculated based on the crown-to-rump length at first trimester ultrasound or a combination of last menstrual period and the fetal biometry when first trimester ultrasound was not available. Preterm birth was classified according to its etiopathogenesis in spontaneous or iatrogenic (when medically induced labour or caesarean section before gestational age 37 weeks) and according to gestational age

thresholds in two groups; less than 34 weeks (moderate and very preterm birth) or 34 weeks or more (late preterm birth).

## Other variables related to the outcome and the exposure

Information about maternal age at the beginning of pregnancy, mothers' height and weight, country of birth, neighbourhood of residence and smoking, alcohol and drug use were obtained from electronic medical records. Maternal chronic health conditions and complications during pregnancy were obtained from the International Classification of Diseases, Ninth Revision, Clinical Modification (ICD-9-CM) codes present in the Minimum Basic Data Set [31] of Spanish National Health System. The maternal country of birth was recoded into 6 regions considering the classification of the World Bank [32]. Deprivation index of the neighbourhood was based on a Catalonia's Government territorial classification used for resource allocation in primary care [33]. The body mass index (BMI) was calculated from the weight and height collected at first follow-up visit if this visit took place before the 14th gestational week. BMI was categorized into underweight ($<20kg/m^2$), normal weight ($20–24.9 kg/m^2$), overweight ($25–29.9 kg/m^2$) and obesity ($>30 kg/m^2$). Pregnant women were asked about smoking, alcohol consumption and drug use at first follow-up visit and any level of consumption was considered. Hypertension and diabetes included the pre-existing conditions and their onset during pregnancy, considering all the ICD-9-CM diagnosis documented. Genitourinary infection included diagnostics of chorioamnionitis, urinary tract infection and vaginitis. Pre-eclampsia and anemia were considered when appeared during pregnancy. Poor prenatal care was recoded when codified as an ICD-9-CM diagnosis.

## Statistical analysis

A directed acyclic graph (DAG) (Fig 1), was created based on previously cited literature to identify confounder variables and intermediate variables [34]. In a preliminary analysis of the database, we selected as potential intermediate variables those associated with preterm birth with a p-

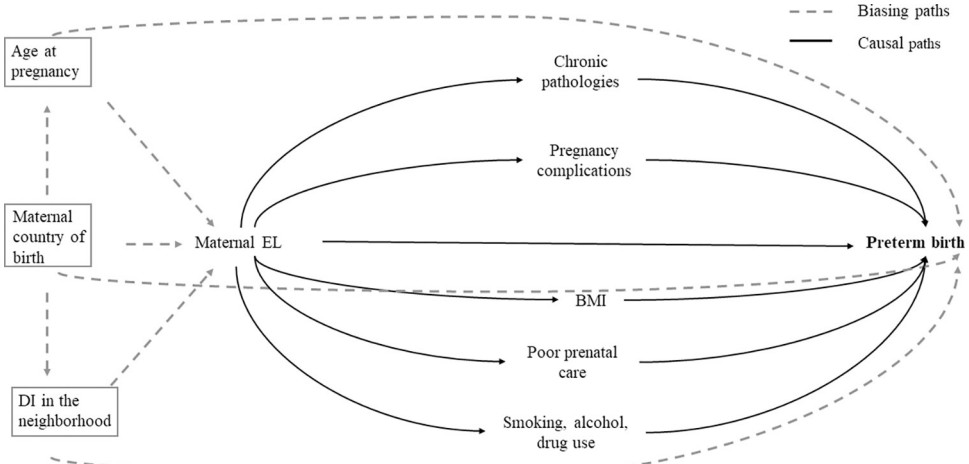

**Fig 1. Direct Acyclic Graph (DAG) of the association between maternal educational level and preterm birth including confounder and intermediate variables DI: Deprivation Index; EL: Educational Level; BMI: Body Mass Index.** The DAG shows some possible causal pathways between educational level and preterm birth. Age at pregnancy, maternal region of birth and DI of the neighbourhood generate biasing pathways since all three variables are associated with both the exposure and the outcome but they are not in the causal pathway. Adjusting for these variables (represented by framing them) block the biasing pathways and maintain the causal pathways of interest.

value <0.20. We performed a univariate analysis to obtain proportions for each categorical variable and the mean and standard deviation for the continuous ones. Relative risk (RR) with 95% confidence intervals (95% CI) of preterm birth as well as RR and 95% CI of the intermediate variables were estimated by constructing bivariate and multivariate modified Poisson regression models with robust error variance. Prevalence ratio (PR) was estimated when the intermediate variable and the exposure were collected at the same time. The intermediate variables were added to an adjusted model separately and all together, examining the role of each mediator in the association between educational level and preterm birth. The percentage of change in the RR was calculated with the formula $[(RR_{without\ the\ mediator} - RR_{with\ the\ mediator})/(1 - RR_{without\ the\ mediator})\ x\ 100]$ [35]. Further, we performed two subanalyses considering the etiology of preterm birth, spontaneous or iatrogenic, and the gestational age thresholds. A complete case analysis was performed as we only had one variable, the BMI, with a small percentage of missing values (5.3%). All the other variables had no missing data. All the analyses were performed using STATA version 13.

## Results

A total of 21437 deliveries took place at our hospital between January 2011 and December 2017. Among these, 11357 belonged to women who lived in the hospital's area. We excluded 261 multiple pregnancies, 49 legal terminations of pregnancy and 580 women with no information in the exposure variable, leaving a study sample of 10467 singleton pregnancies.

Confounder variables identified through the DAG were maternal age at the beginning of pregnancy, maternal region of birth and deprivation index of the neighbourhood as they are related to the exposure and the outcome, but they were not in the causal pathway. We considered BMI, smoking during pregnancy, alcohol consumption, drug use, pregnancy complications such as genitourinary infection, preeclampsia, diabetes, anemia and hypertension and inadequate prenatal care as intermediate variables.

The overall rate of preterm birth was 3.5%, which corresponds to 370 preterm deliveries during the study period. Among those, 231 (62.4%) had a spontaneous etiology and 139 (37.6%) were iatrogenic, 121 (32.7%) were infants born before 34 weeks of gestational age and 249 (67.3%) were born at 34 complete weeks or after. The incidence of preterm birth among the 580 women excluded due to missing data on the exposure variable was 17.1%. One thousand eight hundred twenty-six women (17.4%) had a low educational level, 3607 (34.5%) had a medium educational level and 5034 (48.1%) had a high educational level. The prematurity rate was 4.7% among women with a low educational level, 3.7% among those with a medium level and 3.0% among women with higher education. Less than 20% of women with a high educational level were from low-income countries (Table 1).

Women with low educational level showed an increase of 60% in the risk of preterm birth whereas women with medium educational level appeared to have a non-significant increased risk. The increased RR remained when adjusted for confounder variables (Table 2). In the adjusted model, advanced maternal age (>40) had a significant increase in preterm birth risk compared to women between 25 and 34 years old. We did not find differences in preterm birth risk between women from different countries nor between women living in neighbourhoods with different deprivation index. Overweight and obesity increased the risk of preterm birth as well as smoking, drug use, genitourinary infection, anemia, preeclampsia and diabetes.

Regarding the association between educational level and possible mediators adjusting for confounders (Table 3), we found that women with a low educational level were three times more likely to be obese and almost two times more likely to be overweight compared to high educated women. Medium educational level was also associated with an increased probability of being obese or overweight. The prevalence ratios of smoking and drug use, as well as the RR

**Table 1. Sociodemographic characteristics and pregnancy complications in participant women by educational level.**

| | Educational Level | | |
|---|---|---|---|
| | **Low** | **Medium** | **High** |
| | **n = 1826** | **n = 3607** | **n = 5034** |
| Mean age, y (SD) | 30.26 (6.05) | 31.26 (5.87) | 34.57 (4.30) |
| Maternal region of birth, % | | | |
| Spain | 34.01 | 45.05 | 68.10 |
| High income countries | 2.57 | 4.60 | 12.57 |
| Eastern Europe | 0.77 | 1.25 | 2.48 |
| Central and South America | 19.44 | 27.25 | 9.97 |
| Asia | 32.62 | 17.44 | 5.36 |
| Africa | 10.57 | 4.41 | 1.51 |
| Neighbourhood deprivation index, % | | | |
| Q1 | 11.94 | 16.41 | 28.31 |
| Q2 | 18.07 | 22.59 | 28.45 |
| Q3 | 21.08 | 21.18 | 18.14 |
| Q4 | 19.55 | 16.83 | 12.22 |
| Q5[a] | 29.35 | 22.98 | 12.89 |
| Mean body mass index, kg/m$^2$ (SD) | 24.93 (4.89) | 24.26 (4.38) | 22.68 (3.58) |
| Body mass index, % | | | |
| Underweight | 4.11 | 4.73 | 5.55 |
| Normal | 54.17 | 59.84 | 74.93 |
| Overweight | 27.82 | 24.25 | 14.88 |
| Obesity | 13.91 | 11.19 | 4.65 |
| Diabetes, % | 10.13 | 8.51 | 8.20 |
| Hypertension, % | 1.53 | 2.05 | 1.67 |
| Anemia, % | 5.75 | 4.02 | 3.48 |
| Preeclampsia, % | 3.56 | 3.83 | 2.58 |
| Genitourinary infection, % | 2.14 | 1.47 | 0.97 |
| Smoking during pregnancy, % | 12.92 | 13.25 | 9.38 |
| Alcohol consumption during pregnancy, % | 0.52 | 1.10 | 0.64 |
| Drug use during pregnancy, % | 1.31 | 0.80 | 0.22 |
| Inadequate prenatal care, % | 4.60 | 1.97 | 0.22 |

y: years; SD: Standard Deviation; Q: Quintile

[a] Q5 corresponds to most deprived neighbourhoods

of preeclampsia were higher in women with low and medium educational level. Women with low educational level showed a higher risk of genitourinary infection during pregnancy, whereas women with a medium educational did not.

Table 4 shows the percentage of change in the RR when intermediate variables were added to the model. The overall RR reduction after the inclusion of all intermediate variables was 85.7% in women with a medium educational level and 76.7% in women with a low educational level. The inclusion of all the mediators obscured the association between low educational level and preterm birth (RR 1.14 and 95% CI 0.82, 1.58). BMI accounted for the higher percentage of inequality, as this variable caused the greater RR percentage of change, with 42.9% and 36.7% RR reduction in women with medium and low educational level respectively. When included in the model, smoking and drug use reduced the RR of preterm birth similarly in women of both educational groups. Genitourinary infection had a higher impact on women

**Table 2. RR for the association of preterm birth with educational level, preterm birth with confounder variables and preterm birth with intermediate variables.**

|  | RR | (95% CI) | RR adj | (95% CI) |
|---|---|---|---|---|
| Educational level |  |  |  |  |
| High | 1.00 | (Ref) | 1.00 | (Ref) |
| Medium | 1.23 | (0.98, 1.55) | 1.28 | (1.00, 1.64) |
| Low | 1.57 | (1.21, 2.03) | 1.60 | (1.19, 2.15) |
| Maternal age |  |  |  |  |
| 12–19 | 1.43 | (0.54, 3.78) |  |  |
| 20–24 | 1.19 | (0.80, 1.77) |  |  |
| 25–34 | 1.00 | (Ref) |  |  |
| 35–40 | 1.08 | (0.86, 1.34) |  |  |
| >40 | 1.86 | (1.36, 2.54) |  |  |
| Maternal region of birth |  |  |  |  |
| Spain | 1.00 | (Ref) |  |  |
| High income countries | 0.90 | (0.61, 1.34) |  |  |
| Eastern Europe | 0.61 | (0.23, 1.63) |  |  |
| Central and South America | 0.89 | (0.67, 1.19) |  |  |
| Asia | 1.14 | (0.85, 1.50) |  |  |
| Africa | 1.32 | (0.82, 2.07) |  |  |
| Neighbourhood deprivation index |  |  |  |  |
| Q1 | 1.00 | (Ref) |  |  |
| Q2 | 0.99 | (0.72, 1.35) |  |  |
| Q3 | 1.25 | (0.92, 1.71) |  |  |
| Q4 | 1.14 | (0.81, 1.60) |  |  |
| Q5[a] | 1.23 | (0.90, 1.69) |  |  |
| Body mass index |  |  |  |  |
| Underweight | 1.12 | (0.67, 1.89) | 1.13 | (0.67, 1.89) |
| Normal weight | 1.00 | (Ref) | 1.00 | (Ref) |
| Overweight | 1.63 | (1.27, 2.09) | 1.62 | (1.25, 2.09) |
| Obese | 1.81 | (1.30, 2.53) | 1.80 | (1.28, 2.52) |
| Diabetes | 1.54 | (1.14, 2.07) | 1.44 | (1.06, 1.94) |
| Hypertension | 1.85 | (1.06, 3.23) | 1.70 | (0.97, 2.98) |
| Anemia | 1.93 | (1.33, 2.81) | 1.88 | (1.30, 2.74) |
| Preeclampsia | 6.01 | (4.67, 7.73) | 5.80 | (4.50, 7.47) |
| Genitourinary tract infection | 10.66 | (8.24, 13.78) | 10.75 | (8.32, 13.90) |
| Smoking | 1.51 | (1.16, 1.98) | 1.58 | (1.19, 2.08) |
| Alcohol consumption | 1.56 | (0.60, 4.06) | 1.55 | (0.60, 4.05) |
| Drug use | 4.31 | (2.41, 7.71) | 4.61 | (2.57, 8.27) |
| Inadequate prenatal care | 1.37 | (0.69, 2.72) | 1.27 | (0.63, 2.53) |

RR: Relative Risk; adj: Adjusted for maternal age, maternal region of birth and neighbourhood deprivation index

95%CI: 95% confidence interval; Q: Quintile

[a] Q5 corresponds to most deprived neighbourhoods

with low educational level, accounting for 22.2% of RR inequality. In contrast, the RR of preterm birth was further reduced when preeclampsia was added to the model in medium educated women. The remaining variables produced a percentage of change below 5%.

The detailed results of the subanalyses by etiology and gestational age are available as supporting information. Briefly, women with a low educational level were at higher risk of

**Table 3. Associations between maternal educational level and intermediate variables.**

| | Low Educational Level[a] | | Medium Educational Level[a] | |
|---|---|---|---|---|
| | PR adj | (95% CI) | PR adj | (95% CI) |
| Body mass index | | | | |
| Underweight | 0.88 | (0.66, 1.91) | 1.03 | (0.84, 1.27) |
| Overweight | 1.77 | (1.58, 1.99) | 1.56 | (1.41, 1.71) |
| Obese | 3.15 | (2.58, 3.85) | 2.51 | (2.11, 2.99) |
| Smoking | 2.46 | (2.11, 2.87) | 1.99 | (1.75, 2.25) |
| Alcohol consumption | 3.56 | (1.82, 6.94) | 1.57 | (0.89, 2.78) |
| Drug use | 6.08 | (2.97, 12.4) | 4.08 | (2.06, 8.12) |
| | RR adj | (95% CI) | RR adj | (95% CI) |
| Diabetes | 1.23 | (1.02, 1.48) | 1.12 | (0.97, 1.30) |
| Hypertension | 1.03 | (0.63, 1.68) | 1.29 | (0.90, 1.86) |
| Anemia | 1.42 | (1.07, 1.88) | 1.06 | (0.83, 1.36) |
| Preeclampsia | 1.43 | (1.02, 1.99) | 1.47 | (1.14, 1.91) |
| Genitourinary tract infection | 1.83 | (1.15, 2.93) | 1.27 | (0.84, 1.95) |
| Inadequate prenatal care | 14.58 | (7.11, 29.87) | 7.27 | (3.67, 14.39) |

[a]Ref group: High Educational Level

PR: Prevalence Ratio; RR: Relative Risk; adj: Adjusted for maternal age, maternal region of birth and neighbourhood deprivation index; 95%CI: 95% confidence interval

**Table 4. RR for the associations between maternal educational level and preterm birth before and after including intermediate variables.**

| | RR adj without intermediate variables | RR adj, BMI included | RR adj, smoking included | RR adj, alcohol consumption included | RR adj, drug use included | RR adj, GU infection included |
|---|---|---|---|---|---|---|
| **Educational level** | | | | | | |
| High | 1.00 (Ref) | 1.00 (Ref) | 1.00 (Ref) | 1.00 (Ref) | 1.00 (Ref) | 1.00 (Ref) |
| Medium | 1.28 (1.00, 1.63) | 1.16 (0.89, 1.52) | 1.23 (0.96, 1.57) | 1.28 (1.00, 1.63) | 1.24 (0.97, 1.58) | 1.26 (0.99, 1.61) |
| % Change | -0.0 (Ref) | -42.9 | -17.9 | -0.0 | -14.3 | -7.1 |
| Low | 1.60 (1.19, 2.15) | 1.38 (0.99, 1.92) | 1.51 (1.13, 2.02) | 1.59 (1.18, 2.13) | 1.52 (1.13, 2.03) | 1.49 (1.10, 2.00) |
| % Change | -0.0 (Ref) | -36.7 | -15.0 | -1.7 | -13.3 | -22.2 |
| | RR adj, prenatal care included | RR adj, preeclampsia included | RR adj, diabetes included | RR adj, anemia included | RR adj, hypertension included | RR adj, all variables included |
| High | 1.00 (Ref) | 1.00 (Ref) | 1.00 (Ref) | 1.00 (Ref) | 1.00 (Ref) | 1.00 (Ref) |
| Medium | 1.27 (1.00, 1.63) | 1.21 (0.94, 1.53) | 1.27 (0.99, 1.63) | 1.27 (0.99, 1.63) | 1.27 (1.00, 1.63) | 1.04 (0.80, 1.36) |
| % Change | -3.6 | -25.0 | -3.6 | -3.6 | -3.6 | -85.7 |
| Low | 1.59 (1.18, 2.13) | 1.51 (1.13, 2.03) | 1.58 (1.17, 2.12) | 1.57 (1.17, 2.11) | 1.60 (1.19, 2.15) | 1.14 (0.82, 1.58) |
| % Change | -1.7 | -15.0 | -3.3 | -5.0 | -0.0 | -76.7 |

RR: Relative Risk; adj: Adjusted for maternal age, maternal region of birth and neighbourhood deprivation index:

BMI: Body Mass Index

% Change = [(RR without the mediator–RR with the mediator) / (1- RR without the mediator) x 100]

spontaneous preterm birth and before the 34 weeks of gestation, with greater risks than those observed in the main analysis. The risk of preterm birth after the 34 weeks was similar than the observed for the whole group of preterm infants. For the iatrogenic etiology, a gradient of inequality was also observed, with an increased risk in less educated women that did not reach statistical significance (S1 Table). The distribution and magnitude of risk factors diverged between the subgroups. Overweight and obesity increased the risk of iatrogenic and late preterm birth. Preeclampsia and genitourinary infection were relevant risk factors in all groups, but while preeclampsia was especially prominent in iatrogenic preterm births, genitourinary infection was particularly important in spontaneous preterm birth and preterm infants born before 34 weeks (S1 Table). The mediating role of BMI, smoking, drug use, genitourinary infections and pre-eclampsia in the association between educational level and preterm birth was also observed in the different subgroups, although with different magnitudes. In summary, the factors that accounted for the higher percentage of inequality in women with a low educational level were: infection and smoking in spontaneous preterm births; BMI and preeclampsia in iatrogenic preterm deliveries; the BMI and infection among those born before the 34 weeks of gestational age and the BMI, smoking and preeclampsia among those born after 34 weeks (S2 Table).

## Discussion

### Main findings and interpretation

First, several factors may explain the small preterm birth incidence observed in our cohort, which, although low, shows the usual proportion in terms of etiology and gestational age thresholds [36]. On the one hand, although preterm birth rates in high complexity hospitals are usually above population rates, the selection of participants in this study have attempted to represent the general pregnant women population. On the other hand, those women with missing data about the educational level, which were excluded from the analysis, represented a high preterm birth burden. Our interpretation is that these women did not have this information collected because they did not attend the regular follow-up visits, which could be related to reduced accessibility to health system, poor health literacy level and low socioeconomic status. Therefore, inequalities in our preterm birth rates could be greater than those reported in this study. Finally, the Hospital Clínic is a high complexity centre where early detection and prevention practices are part of the standard of care clinical management of pregnancy, regardless of their obstetric risk. In spite of this, educational disparities are still present in Hospital Clínic's perinatal outcomes and we consider these results could be applied to other populations with similar social disparities and healthcare resources.

Women with low educational level have a significant increased risk of preterm birth in comparison to women with a high educational level. Instead, women with medium educational level have a non-statistically significant increased risk. The increased risk seems to be explained by intermediate variables related to both exposure and outcome. The BMI, smoking, drug use, genitourinary infections and preeclampsia were the variables that reduced the RR of preterm birth the most when included in the model.

Women with low and medium educational level are more likely to have overweight and obesity in our population, even when adjusted for maternal age, region of birth and deprivation index of the neighbourhood. These inequalities in the social distribution of overweight and obesity have been observed in many studies performed in middle and high-income countries, especially in women [24, 25]. At the same time, we observed an increased risk of having a preterm birth in women with obesity and overweight. The role of BMI as a relevant mediator between socioeconomic level and health outcomes such as type 2 diabetes [37] and

cardiovascular disease [38] has been established. BMI is considered a contributing factor in the higher risk of preterm birth in Dutch women with a low level of education compared to women with a high educational level. In contrast, it does not seem to be involved in the inequalities observed between women with medium and high educational level [22]. However, in our study BMI accounted for an important percentage of inequality in both categories, low and medium educational level, probably due to the high prevalence of overweight and obesity observed in both groups.

Smoking was unequally distributed among participants in this study. Women with low and medium educational level had a higher PR of smoking and at the same time, women who smoked during pregnancy showed an increased risk of preterm birth. Drug use followed the same pattern, although its prevalence was lower. Both harmful health behaviours partially explained the RR inequalities observed. Our findings conform with results in other European cohort studies [21, 39], which concluded that smoking partly explains educational disparities in preterm birth. Moreover, Gavin et al [40] found that social disadvantage was associated with substance abuse, which contributed to worse maternal health and poor birth outcomes.

The onset of preeclampsia and genitourinary infection during pregnancy produced the greatest increase in preterm birth RR in this study. Both factors had a higher incidence in women with a low educational level. In contrast, women with a medium educational level had a higher risk of preeclampsia without an increase of genitourinary infection when compared to women with high educational level.

A cohort-study conducted in California observed a mediation role of preeclampsia for the association between socioeconomic level, based on educational level and insurance status, and preterm birth: socioeconomically advantaged white women had a longer gestational period partly explained by a reduction in the risk of preeclampsia [41], but this mediated effect was not found in black women. The present study showed a reduction in RR when preeclampsia was included in an adjusted model for mother's region of birth, but we did not consider the participants' ethnicity. Genitourinary infection was considered a possible contributor to preterm birth inequality early in literature [42] although its mediation role is unclear. Socioeconomically disadvantaged women are more exposed to mental health problems and stressful life events, whose biological response is related to systemic inflammation and may increase the risk of genitourinary infections during pregnancy, contributing to inequalities in preterm birth [42, 43]. Hypertension and diabetes did not increase the risk of preterm birth in our population in contrast to previous studies [6, 7, 9, 11]. Grouping the chronic condition and their onset during pregnancy could explain these results in our analysis.

Unlike other studies [4, 44, 45], women who referred alcohol consumption in the first follow-up visit did not show an increased risk of preterm birth in our study. These differences could be explained by the small number of women consuming alcohol in our population and the lack consumption level data. Finally, we observed a marginal reduction when anemia and inadequate prenatal care were included in the model although both factors increase the risk of preterm birth. Nevertheless, the role of this factors can be hampered by the small sample size of women with these characteristics in the database.

Regarding the subanalyses results, statistically significant educational disparities were found in spontaneous preterm birth and both gestational age groups. Although we observed a gradient of inequality for iatrogenic preterm birth, it did not reach statistical significance, consistently with prior findings [46], even when adjusted by maternal age, which has been pointed as a possible explanation of the increase in preterm birth rates among socioeconomically advantaged women [47]. The BMI accounted for an important percentage of inequality in infants born before and after the 34 weeks, as well as it reduced the non-significant RR increase in iatrogenic preterm birth, but their role was less important in spontaneous preterm birth.

This result is in consonance with the findings of Brink et al [46], who observed a lower prevalence of obesity in women who had spontaneous preterm births compared with iatrogenic. Infection was especially relevant in inequalities observed in spontaneous preterm birth and infants born before the 34 weeks, consistently with prior results regarding the role of infection in deprivation and spontaneous and very preterm birth [48]. Finally, it is worth mentioning that the etiopathogenesis classification of preterm birth (spontaneous vs iatrogenic) has proven to be insufficient to explain the complexity of preterm birth syndrome [30, 49] and a new classification has been proposed, based on twelve preterm birth phenotypes [50]. Using this classification in future research could lead to a better understanding of disparities in the etiology of preterm birth.

## Strengths and limitations

The main strengths of this study are the large population and the availability of individual-level data about educational level, used as a proxy indicator of socioeconomic level. Data related to anthropometric measures, substance use, chronic health conditions and pregnancy complications was available and used in the analysis to assess their role in preterm birth inequalities. In order to reduce the possible selection bias in hospital-based studies, women with a different reference health facility who were referred to our high complexity hospital due to high obstetric risk were identified and excluded.

Despite data was collected prospectively, cross-sectional data was also included: maternal educational level, confounder variables and some intermediate variables such as BMI, smoking, alcohol consumption and drug use were collected once, during the first follow-up visit. This may limit the interpretation of the results as causality cannot not be established between educational level and these factors. Although maternal region of birth was considered a confounder variable in the analysis, we did not have available data about ethnicity nor about the time since their arrival to the country. Maternal occupation and household income may play a confounder role in the association between maternal education and preterm birth, but they were not available for analysis. Some intermediate variables were considered only when the ICD-9-CM diagnose was assigned by a medical practitioner. Therefore, overrepresentation of most severe cases of some conditions such as preeclampsia or genitourinary infection should be contemplated. Finally, due to the lack of this variable in the database, it has not been possible to assess the mediator role of assisted reproductive techniques, which are more prevalent in women with a high socioeconomic status [51] and have been established as a risk factor of preterm birth [52].

## Implications

Findings of this study suggest that low educational level increases the risk of preterm birth and that some medical conditions and harmful health behaviours more prevalent in social disadvantage women could explain this inequality. The health literacy level, which has been found to be insufficient in almost 60% of Spanish population [53], may explain the higher prevalence of harmful health behaviours and worse medical conditions in women with lower educational level. The implementation of preventive and health promotion strategies, including the improvement of health literacy in our population, may have an impact on these mediator mechanisms, decreasing the preterm birth inequalities [53]. Moreover, the identification of most relevant preterm birth risk factors for specific populations may help clinicians when addressing individual patient risk [54].

Firstly, the prevalence of maternal smoking is high in our population, around 13% in women with low and medium educational level, although it does not reach the 20% observed at national level in Spain [55]. Moreover, it is well known that low health literacy is associated with maternal

smoking [40]. Thus, national and regional smoking cessation programs should be improved and their coverage extended, particularly addressing women with low health literacy.

Furthermore, high risk of preeclampsia and genitourinary infection in women with a lower educational level could be decreased through interventions that promote early detection. Whitney et al [56] demonstrated that pregnant women who were exposed to an educational tool about preeclampsia symptoms achieved a higher knowledge of this disease, suggesting that the improvement of health literacy in pregnant women may increase early diagnosis of preeclampsia and reduce its consequences. More research about health literacy inequalities among pregnant women may help to clarify the mechanisms of inequality in preterm birth and other pregnancy outcomes.

Regarding the role of BMI, we observed a concerning prevalence of high BMI in women with a low and medium levels of education. Although regional and national clinical guidelines include nutritional counselling for pregnant women, there are no available information about the coverage and outcomes of this health promotion activity [57]. On one hand, health literacy might be a tool to increase patient information on nutrition and physical activity, before and during pregnancy, contributing to decrease overweight effects on pregnancy outcomes [58]. On the other hand, a greater effort must be done by policy makers to promote healthy environments and to prevent overweight and obesity in young populations. Lastly, women with a low educational level and socioeconomic level have worse living conditions, less family and institutional support and are more exposed to stressful situations in comparison to women with a higher socioeconomic level [42]. These factors lead to worse mental health conditions that have been associated with a higher preterm birth risk [59].

In conclusion, we found maternal educational inequalities in preterm birth risk in the study population. These inequalities are explained by factors such as overweight and obesity, smoking, drug use, preeclampsia and genitourinary infections. A better understanding about preterm birth inequalities' etiology may help clinical practitioners, public health officers and policy makers to find strategies for prevention of preterm deliveries and reduction of perinatal health inequalities.

## Supporting information

**S1 Table. RR of spontaneous preterm birth, iatrogenic preterm birth, preterm birth before 34 weeks of gestational age and preterm birth from 34 weeks of gestational age according to educational level, maternal health conditions and health-related behaviours.** RR: Relative Risk; adj: Adjusted for maternal age, maternal region of birth and neighbourhood deprivation index; 95%CI: 95% confidence interval.
(PDF)

**S2 Table. RR for the associations between maternal educational level and different groups of preterm birth (spontaneous, iatrogenic, less than 34 weeks and 34 weeks or more) before and after including intermediate variables and calculation of the percentage of change in the RR.** RR: Relative Risk; adj: Adjusted for maternal age, maternal region of birth and neighbourhood deprivation index; BMI: body mass index; % change = [(RR without the mediator–RR with the mediator) / (1- RR without the mediator) x 100].
(PDF)

## Acknowledgments

This study could not have been carried out without all professionals who design and implemented the systematic data collection in our hospital and without the everyday effort of

obstetricians, nurses and other health care providers working in the maternal-fetal medicine department of the Hospital Clínic of Barcelona. We also thank to all those women who trusted our hospital for their pregnancy and childbirth.

## Author Contributions

**Conceptualization:** Laura Granés, Montse Palacio, Laura De la Torre, Anna Llupià.

**Data curation:** Laura Granés.

**Formal analysis:** Laura Granés, Isabel Torà-Rocamora.

**Investigation:** Montse Palacio.

**Methodology:** Laura Granés, Isabel Torà-Rocamora, Laura De la Torre, Anna Llupià.

**Supervision:** Anna Llupià.

**Writing – original draft:** Laura Granés, Anna Llupià.

**Writing – review & editing:** Laura Granés, Isabel Torà-Rocamora, Montse Palacio, Laura De la Torre.

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
