## [Decision Letter · Decision Letter 0]

12 Dec 2022

PONE-D-22-22954Maternal educational level and preterm birth: Exploring inequalities in a hospital-based cohort studyPLOS ONE

Dear Dr. Granés,

Thank you for submitting your manuscript to PLOS ONE. After careful consideration, we feel that it has merit but does not fully meet PLOS ONE’s publication criteria as it currently stands. Therefore, we invite you to submit a revised version of the manuscript that addresses the points raised during the review process.

The manuscript and the reviewers’ comments were carefully evaluated. The Reviewers appreciated the manuscript; however, they highlighted some minor points that require revisions before considering the manuscript for publication. Suggested improvements and changes are in detail reported in the Reviewers’ comments. 

We look forward to receiving your revised manuscript.

Kind regards,

Simone Garzon

Academic Editor

PLOS ONE

Journal Requirements:

Reviewers' comments:

Reviewer's Responses to Questions

**Comments to the Author**

1. Is the manuscript technically sound, and do the data support the conclusions?

Reviewer #1: Yes

Reviewer #2: Yes

2. Has the statistical analysis been performed appropriately and rigorously? 

Reviewer #1: Yes

Reviewer #2: I Don't Know

3. Have the authors made all data underlying the findings in their manuscript fully available?

Reviewer #1: Yes

Reviewer #2: Yes

4. Is the manuscript presented in an intelligible fashion and written in standard English?

Reviewer #1: Yes

Reviewer #2: Yes

5. Review Comments to the Author

Reviewer #1: The authors demonstrated that educational status may be indeed work as risk factor of preterm birth. This is due to the effects over different covariates including BMI, smoking drugs and genitourinary infections. This is not novel nor surpising but the methods is valid and results are sound. I have a good impression of this work and I'm providing some suggestions for improving the paper before publication:

A recent study established a generalized methodology with machine learning for building-up an evidence-based holistic risk assessment for PTB to be used in clinical practice. This evidence should be cited (ref 1) along with the concept that selection of predictive risk facctors is essential along with a hierarchical procedure.

A further limitation is lack of etiological differentiation of PTB: iatrogenic vs spontaneous. This etiological dichotomy is particularly prominent in IVF pregnancies which are indeed a high-risk group for preterm birth (ref 2). I reccomend citing this concept and considering this variable for their study. If this is not possible it should be mentioned as a limitation along with the appropriate referencing.

recent work showed that phenotypic classification of PTB may provide a better understanding of the etiologic factors and mechanisms associated with preterm birth than continuing to consider it an exclusively time-based entity. (ref 3). This fits with my comment number 2 related to etiology. Do they believe that the model would work better truing to predict different and specific PTB phenotypes/etiology? add a comment on that.

The same comments related to etiology (dichotomic) or phenotype (multiple classes) may be applied to gestational age tresholds. Did they try to assess PTB<34 or 28 weeks? What are the results? And the implications? The study would be much stronger if they try to look at different GA tresholds.

References

1. Della Rosa PA, et al. A hierarchical procedure to select intrauterine and extrauterine factors for methodological validation of preterm birth risk estimation. BMC Pregnancy Childbirth. 2021 Apr 16;21(1):306. doi: 10.1186/s12884-021-03654-3. PMID: 33863296; PMCID: PMC8052693.

2. Cavoretto P, et al. Risk of spontaneous preterm birth in singleton pregnancies conceived after IVF/ICSI treatment: meta-analysis of cohort studies. Ultrasound Obstet Gynecol. 2018 Jan;51(1):43-53. doi: 10.1002/uog.18930. PMID: 29114987.

3. Villar J, et al. JAMA Pediatr. 2021 May 1;175(5):483-493. doi: 10.1001/jamapediatrics.2020.6087. PMID: 33646288; PMCID: PMC7922239.

Reviewer #2: Thank you for this interesting manuscript that provides insight how the effect of sociodemographic factors influence birth outcomes. I have some minor questions

1) Could you elaborate on the fact of low incidence of preterm birth in your cohort i.e are results applicable in another population?

2) The authors state that all types of preterm birth was included and no further distinction on onset or type was performed. I think it could actually differ between spontaneous and induced onset how much educational level matters as a factor and also how the mediating factors differ. Could you elaborate on this in analysis or if no data is available mention as a limitation if you agree.

3) Different approaches (statistical) of mediation could be used. I understand that this is based on logistic regression and a sort of attributable factor calculation. Did the authors consider to conduct other type of mediation calculation for example SEM analysis, and how does this analysis differ in the statistical epidemiological approach of the method used here.

6. PLOS authors have the option to publish the peer review history of their article (what does this mean?). If published, this will include your full peer review and any attached files.

Reviewer #1: No

Reviewer #2: No

---

## [Author Response · Author response to Decision Letter 0]

14 Feb 2023

Reviewer #1

The authors demonstrated that educational status may be indeed work as risk factor of preterm birth. This is due to the effects over different covariates including BMI, smoking drugs and genitourinary infections. This is not novel nor surprising but the methods is valid and results are sound. I have a good impression of this work and I'm providing some suggestions for improving the paper before publication:

First of all, the research team would like to sincerely thank you for your time and effort reading and evaluating our work, for your positive assessment and your very interesting suggestions. Your review has allowed us to get to knowledge of recent and relevant bibliography, to add important limitations, to include two subanalyses of the data considering the preterm birth etiology and gestational age thresholds. We consider that all these changes have definitely improved the manuscript. 

*All line numbers refer to the document “Revised Manuscript Track Changes”

1. A recent study established a generalized methodology with machine learning for building-up an evidence-based holistic risk assessment for PTB to be used in clinical practice. This evidence should be cited (ref 1) along with the concept that selection of predictive risk factors is essential along with a hierarchical procedure.

Thank you very much for suggesting this study. We found its results very relevant. It has allowed us to reflect in depth on the usefulness of identifying risk factors in clinical practice. We added a sentence about the study in the discussion section.

Discussion section, line 359: “Moreover, the identification of most relevant preterm birth risk factors for specific populations may help clinicians when addressing individual patient risk [Ref].”

Ref: Della Rosa PA, Miglioli C, Caglioni M, Tiberio F, Mosser KHH, Vignotto E, et al. A hierarchical procedure to select intrauterine and extrauterine factors for methodological validation of preterm birth risk estimation. BMC Pregnancy Childbirth. 2021;21(1):1–17. doi: 10.1186/S12884-021-03654-3.

2. A further limitation is lack of etiological differentiation of PTB: iatrogenic vs spontaneous. This etiological dichotomy is particularly prominent in IVF pregnancies which are indeed a high-risk group for preterm birth (ref 2). I recommend citing this concept and considering this variable for their study. If this is not possible it should be mentioned as a limitation along with the appropriate referencing.

Regarding the etiological differentiation, we address this issue in the following points (3&4). Regarding the IVF, we fully agree that this is a relevant issue as IVF are more prevalent in women with high socioeconomic status and they are a risk factor of preterm birth. Unfortunately, we do not have the variable informing about “IVF” or “assisted reproductive technique” in our database. We have added the limitation with two relevant citations in “Strengths and limitations” section. 

Strengths and limitations section, line 347: “Finally, due to the lack of this variable in the database, it has not been possible to assess the mediator role of assisted reproductive techniques, which are more prevalent in women with a high socioeconomic status [Ref1] and have been established as a risk factor of preterm birth [Ref2].”

Ref1: Sauer M V. Reproduction at an advanced maternal age and maternal health. Fertil Steril.

2015;103(5):1136–43. doi: 10.1016/J.FERTNSTERT.2015.03.004.

Ref2: Pandey S, Shetty A, Hamilton M, Bhattacharya S, Maheshwari A. Obstetric and perinatal

outcomes in singleton pregnancies resulting from IVF/ICSI: a systematic review and meta

analysis. Hum Reprod Update. 2012;18(5):485–503. doi:

3. Recent work showed that phenotypic classification of PTB may provide a better understanding of the etiologic factors and mechanisms associated with preterm birth than continuing to consider it an exclusively time-based entity. (ref 3). This fits with my comment number 2 related to etiology. Do they believe that the model would work better trying to predict different and specific PTB phenotypes/etiology? add a comment on that.

4. The same comments related to etiology (dichotomic) or phenotype (multiple classes) may be applied to gestational age thresholds. Did they try to assess PTB<34 or 28 weeks? What are the results? And the implications? The study would be much stronger if they try to look at different GA thresholds.

Answers to points 3 & 4: 

Initially, we aimed to use the etiology of preterm birth (spontaneous vs. iatrogenic) in our analysis, but considering some experts opinion regarding the limitations of the dichotomous classification of preterm birth (works cited in our article; Kramer et al, 2012 & Goldenberg et al, 2011) we decided not to use it. Even so, we have this variable available, so after reading your comments and considering that both reviewers agreed that it would improve de manuscript, we considered appropriate to perform a subanalysis according to the dichotomic etiology and to point out that recent literature indicates that this classification is somewhat obsolete and citing the work in relation to the identification of more than two phenotypes of preterm birth. Further, we have added a subanalysis according to gestational age thresholds <34 weeks and 34 weeks or more (considering the sample size of these two groups, which is enough to perform the analysis). We have added the complete results of the subanalyses as a supplementary material S1 & S2 Tables, and the following comments in the main article: 

Methods

• Line 97 “Preterm birth was classified according to its etiopathogenesis in spontaneous or iatrogenic (when medically induced labour or caesarean section before gestational age 37 weeks) and according to gestational age thresholds in two groups; less than 34 weeks (moderate and very preterm birth) or 34 weeks or more (late preterm birth).”

• Line 135 “Further, we performed two subanalyses considering the aetiology of preterm birth, spontaneous or iatrogenic, and the gestational age threshold.”

Results

• Line 162 “Among those, 231 (62.4%) had a spontaneous etiology and 139 (37.6%) were iatrogenic and 121 (32.7%) were infants born before 34 weeks of gestational age and 249 (67.3%) were born at 34 complete weeks or after.”

• Line 220 “The detailed results of the subanalyses by etiology and gestational age are available as supporting information. Briefly, women with a low educational level were at higher risk of spontaneous preterm birth and before the 34 weeks of gestation, with greater risks than those observed in the main analysis. The risk of preterm birth after the 34 week was similar than the observed for the whole group of preterm infants. For the iatrogenic etiology, a gradient of inequality was also observed, with an increased risk in less educated women that did not reach the statistical significance (S1 Table). The distribution and magnitude of risk factors diverged between the subgroups. Overweight and obesity increased the risk of iatrogenic and late preterm birth. Preeclampsia and genitourinary infection were relevant risk factors in all groups, but while preeclampsia was especially prominent in iatrogenic preterm births, genitourinary infection was particularly important in spontaneous preterm birth and preterm infants born before 34 weeks (S1 Table). The mediating role of BMI, smoking, drug use, genitourinary infections and pre-eclampsia in the association between educational level and preterm birth was also observed in the different subgroups, although with different magnitudes. In summary, the factors that accounted for the higher percentage of inequality in women with a low educational level were: infection and smoking in spontaneous preterm births; BMI and preeclampsia in iatrogenic preterm deliveries; the BMI and infection among those born before the 34 weeks of gestational age and the BMI, smoking and preeclampsia among those born after 34 weeks (S2 Table).”

Discussion

• Line 310 “Regarding the subanalyses results, statistically significant educational disparities were found in spontaneous preterm birth and both gestational age groups. Although we observed a gradient of inequality for iatrogenic preterm birth, it did not reach the statistical significance consistently with prior findings [Ref1], even when adjusted by maternal age, which has been pointed as a possible explanation of the increase in preterm birth rates among socioeconomically advantaged women [Ref2]. The BMI accounted for an important percentage of inequality in infants born before and after the 34 weeks, as well as it reduced the RR observed in iatrogenic preterm birth, but their role was less important in spontaneous preterm birth. This result is in consonance with the findings of Brink et al [Ref1], who observed a lower prevalence of obesity in women who had spontaneous preterm births compared with iatrogenic. Infection was especially relevant in inequalities observed in spontaneous preterm birth and infants born before the 34 weeks, consistently with prior results regarding the role of infection in deprivation and spontaneous and very preterm birth [Ref3]. Finally, it is worth mentioning that the etiopathogenesis classification of preterm birth (spontaneous vs iatrogenic) has proven to be insufficient to explain the complexity of preterm birth syndrome [Ref4, Ref5] and a new classification has been proposed, based on twelve preterm birth phenotypes[Ref6] Using this classification in future research could lead to a better understanding of disparities in preterm birth according to its etiology mechanisms.

Ref1: Brink LT, Nel DG, Hall DR, Odendaal HJ. Association of socioeconomic status and clinical and demographic conditions with the prevalence of preterm birth. Int J Gynaecol Obstet. 2020;149(3):359. doi: 10.1002/IJGO.13143.

Ref2: Joseph K, Fahey J, Shankardass K, Allen VM, O’campo P, Dodds L, et al. Effects of socioeconomic position and clinical risk factors on spontaneous and iatrogenic preterm birth. BMC Pregnancy Childbirth. 2014; 14(117). doi: 10.1186/1471-2393-14-117

Ref3: Smith LK, Draper ES, Manktelow BN, Field DJ. Deprivation and infection among spontaneous very preterm births. Obstet Gynecol. 2007;110(2 I):325–9. doi: 10.1097/01.AOG.0000270158.57566.2F.

Ref4: Kramer MS, Papageorghiou A, Culhane J, Bhutta Z, Goldenberg RL, Gravett M, et al. Challenges in defining and classifying the preterm birth syndrome. Am J Obstet Gynecol. 2012;206(2):108–12. doi: 10.1016/j.ajog.2011.10.864.

Ref5: Goldenberg RL, Gravett MG, Iams J, Papageorghiou AT, Waller SA, Kramer M, et al. The preterm birth syndrome: issues to consider in creating a classification system. Am J Obstet Gynecol. 2012;206(2):113–8. doi: 10.1016/j.ajog.2011.10.865

Ref6. Barros FC, Papageorghiou AT, Victora CG, Noble JA, Pang R, Iams J, et al. The distribution of clinical phenotypes of preterm birth syndrome: implications for prevention. JAMA Pediatr. 2015;169(3):220–9. doi: 10.1001/JAMAPEDIATRICS.2014.3040.

Reviewer #2

Thank you for this interesting manuscript that provides insight how the effect of sociodemographic factors influences birth outcomes. I have some minor questions. 

First of all, the research team would like to sincerely thank you for your time and effort reading and evaluating our work, for your positive assessment and your very interesting suggestions that have improved the manuscript. Your review has allowed us to think more deeply about some questions, such as the low prevalence of preterm birth in our cohort (which is definitely an issue we must have addressed) and the need to include a subanalyses considering the etiology of preterm birth. 

*All line numbers refer to the document “Revised Manuscript Track Changes”

1. Could you elaborate on the fact of low incidence of preterm birth in your cohort i.e are results applicable in another population? 

Thank you for this first comment, it is a very relevant point that we had missed. The overall preterm birth rate in our cohort is 3.5%, which is indeed lower than the rate in Barcelona (6.5%) and in Spain (7%). 

There are several factors that could explain this low prevalence. 

Although tertiary hospitals usually have higher rates of preterm birth, we excluded from our analysis all these women that have been referred to our hospital because of their high obstetric risk: The Hospital Clinic of Barcelona is a public high complexity hospital that provides care to two different populations. On the one hand, all women living in four districts of Barcelona city are assigned to the Hospital Clínic for their pregnancy follow-up, regardless their obstetric risk. On the other hand, Hospital Clínic assists women referred due to high-risk pregnancies from other hospitals in semi-urban or rural areas. In order to minimise the selection bias inherent in hospital-based studies, for this study we only included women of the first group, who lived in Barcelona and were a priori assigned to our hospital. We have added this explanation in the methods section for a better understanding of the participants’ selection. 

Considering the above, this low incidence of preterm birth is probably explained by the fact that women without a specific high risk are followed in a high complexity centre where early detection and prevention practices are included in the usual clinical management of pregnancy. 

Moreover, we have to add that the prevalence of preterm birth would be higher (4.25%) if we had included those women without information in the exposure variable. Among the 580 women who did not have information about the educational level, the rate of preterm birth was 17.1%. Our interpretation is that these women did not attend regular follow-up visits (where the variable educational level is collected), which is a sign of reduced accessibility to the health system, poor health literacy and low socioeconomic level. We have also included this important information in the manuscript. 

So, despite of the high standards in clinical practices, socioeconomic disparities are still present in our hospital (and probably they are higher than reported here) and, therefore, we consider that these results are applicable to other populations with similar social disparities, even though they have higher rates of preterm birth. 

We have included these explanations and interpretations in the following sections: 

Methods, line 74 “The Hospital Clinic of Barcelona is a public high complexity hospital that provides care to two different populations. On the one hand, all women living in four districts of Barcelona city are assigned to the Hospital Clínic for their pregnancy follow-up, regardless of their obstetric risk. On the other hand, Hospital Clínic assists women referred due to high-risk pregnancies from other hospitals in semi-urban or rural areas. In order to minimise the selection bias inherent in hospital-based studies, for this study we only included women of the first group, who lived in Barcelona and were a priori assigned to our hospital.” 

Results, line 164 “The incidence of preterm birth among the 580 women excluded due to missing data on the exposure variable was 17.1%.”

Discussion, line 241: “First, several factors may explain the small preterm birth incidence observed in our cohort, which, although low, shows the usual proportion in terms of etiology and gestational age thresholds [Ref]. On the one hand, although preterm birth rates in high complexity hospitals are usually above population rates, the selection of participants in this study have attempt to represent general pregnant women population. On the other hand, those women with missing data on the educational level, which were excluded from the analysis, represented a high preterm birth burden. Our interpretation is that these women did not have this information collected because they did not attend the regular follow-up visits, which could be related to reduced accessibility to the health system, poor health literacy level and low socioeconomic status. Therefore, inequalities in our preterm birth rates could be greater than those reported in this study. Finally, the Hospital Clínic is a high complexity centre where early detection and prevention practices are part of the standard of care clinical management of pregnancy, regardless of their obstetric risk. In spite of this, educational disparities are still present in Hospital Clinic’s perinatal outcomes and we consider these results could be applied to other populations with similar social disparities and healthcare resources.”

Ref: Goldenberg RL, Culhane JF, Iams JD, Romero R. Epidemiology and causes of preterm birth. Lancet. 2008;371(9606):75–84. doi: 10.1016/S0140-6736(08)60074-4

2. The authors state that all types of preterm birth were included and no further distinction on onset or type was performed. I think it could actually differ between spontaneous and induced onset how much educational level matters as a factor and also how the mediating factors differ. Could you elaborate on this in analysis or if no data is available mention as a limitation if you agree.

Initially, we aimed to use the etiology of preterm birth (spontaneous vs. iatrogenic) in our analysis, but considering some experts opinion regarding the limitations of the dichotomous classification of preterm birth (works cited in our article; Kramer et al, 2012 & Goldenberg et al, 2011) we decided not to use it. Even so, we have this variable available, so after reading your comments and considering that both reviewers agreed that it would improve de manuscript, we considered appropriate to perform a subanalysis according to the dichotomic etiology and to point out that recent literature indicates that this classification is somewhat obsolete and citing the work in relation to the identification of more than two phenotypes of preterm birth. Further, we have added a subanalysis according to gestational age thresholds <34 weeks and 34 weeks or more (considering the sample size of these two groups, which is enough to perform the analysis). We have added the complete results of the subanalyses as a supplementary material S1 & S2 Tables, and the following comments in the main article: 

Methods

• Line 97 “Preterm birth was classified according to its etiopathogenesis in spontaneous or iatrogenic (when medically induced labour or caesarean section before gestational age 37 weeks) and according to gestational age thresholds in two groups; less than 34 weeks (moderate and very preterm birth) or 34 weeks or more (late preterm birth).”

• Line 135 “Further, we performed two subanalyses considering the aetiology of preterm birth, spontaneous or iatrogenic, and the gestational age threshold.”

Results

• Line 162 “Among those, 231 (62.4%) had a spontaneous etiology and 139 (37.6%) were iatrogenic and 121 (32.7%) were infants born before 34 weeks of gestational age and 249 (67.3%) were born at 34 complete weeks or after.”

• Line 220 “The detailed results of the subanalyses by etiology and gestational age are available as supporting information. Briefly, women with a low educational level were at higher risk of spontaneous preterm birth and before the 34 weeks of gestation, with greater risks than those observed in the main analysis. The risk of preterm birth after the 34 week was similar than the observed for the whole group of preterm infants. For the iatrogenic etiology, a gradient of inequality was also observed, with an increased risk in less educated women that did not reach the statistical significance (S1 Table). The distribution and magnitude of risk factors diverged between the subgroups. Overweight and obesity increased the risk of iatrogenic and late preterm birth. Preeclampsia and genitourinary infection were relevant risk factors in all groups, but while preeclampsia was especially prominent in iatrogenic preterm births, genitourinary infection was particularly important in spontaneous preterm birth and preterm infants born before 34 weeks (S1 Table). The mediating role of BMI, smoking, drug use, genitourinary infections and pre-eclampsia in the association between educational level and preterm birth was also observed in the different subgroups, although with different magnitudes. In summary, the factors that accounted for the higher percentage of inequality in women with a low educational level were: infection and smoking in spontaneous preterm births; BMI and preeclampsia in iatrogenic preterm deliveries; the BMI and infection among those born before the 34 weeks of gestational age and the BMI, smoking and preeclampsia among those born after 34 weeks (S2 Table).”

Discussion

• Line 310 “Regarding the subanalyses results, statistically significant educational disparities were found in spontaneous preterm birth and both gestational age groups. Although we observed a gradient of inequality for iatrogenic preterm birth, it did not reach the statistical significance consistently with prior findings [Ref1], even when adjusted by maternal age, which has been pointed as a possible explanation of the increase in preterm birth rates among socioeconomically advantaged women [Ref2]. The BMI accounted for an important percentage of inequality in infants born before and after the 34 weeks, as well as it reduced the RR observed in iatrogenic preterm birth, but their role was less important in spontaneous preterm birth. This result is in consonance with the findings of Brink et al [Ref1], who observed a lower prevalence of obesity in women who had spontaneous preterm births compared with iatrogenic. Infection was especially relevant in inequalities observed in spontaneous preterm birth and infants born before the 34 weeks, consistently with prior results regarding the role of infection in deprivation and spontaneous and very preterm birth [Ref3]. Finally, it is worth mentioning that the etiopathogenesis classification of preterm birth (spontaneous vs iatrogenic) has proven to be insufficient to explain the complexity of preterm birth syndrome [Ref4, Ref5] and a new classification has been proposed, based on twelve preterm birth phenotypes [Ref6]. Using this classification in future research could lead to a better understanding of disparities in preterm birth according to its etiology mechanisms.

Ref1: Brink LT, Nel DG, Hall DR, Odendaal HJ. Association of socioeconomic status and clinical and demographic conditions with the prevalence of preterm birth. Int J Gynaecol Obstet. 2020;149(3):359. doi: 10.1002/IJGO.13143.

Ref2: Joseph K, Fahey J, Shankardass K, Allen VM, O’campo P, Dodds L, et al. Effects of socioeconomic position and clinical risk factors on spontaneous and iatrogenic preterm birth. BMC Pregnancy Childbirth. 2014; 14(117). doi: 10.1186/1471-2393-14-117

Ref3: Smith LK, Draper ES, Manktelow BN, Field DJ. Deprivation and infection among spontaneous very preterm births. Obstet Gynecol. 2007;110(2 I):325–9. doi: 10.1097/01.AOG.0000270158.57566.2F.

Ref4: Kramer MS, Papageorghiou A, Culhane J, Bhutta Z, Goldenberg RL, Gravett M, et al. Challenges in defining and classifying the preterm birth syndrome. Am J Obstet Gynecol. 2012;206(2):108–12. doi: 10.1016/j.ajog.2011.10.864.

Ref5: Goldenberg RL, Gravett MG, Iams J, Papageorghiou AT, Waller SA, Kramer M, et al. The preterm birth syndrome: issues to consider in creating a classification system. Am J Obstet Gynecol. 2012;206(2):113–8. doi: 10.1016/j.ajog.2011.10.865

Ref6. Barros FC, Papageorghiou AT, Victora CG, Noble JA, Pang R, Iams J, et al. The distribution of clinical phenotypes of preterm birth syndrome: implications for prevention. JAMA Pediatr. 2015;169(3):220–9. doi: 10.1001/JAMAPEDIATRICS.2014.3040.

3. Different approaches (statistical) of mediation could be used. I understand that this is based on logistic regression and a sort of attributable factor calculation. Did the authors consider to conduct other type of mediation calculation for example SEM analysis, and how does this analysis differ in the statistical epidemiological approach of the method used here.

Thank you for this comment. From the beginning we proposed this type of analysis, inspired by other articles on social inequalities that used this method, which we find easy to interpret, in the sense of being able to observe how the relative risk changes when adding the different risk factors and to understand the underlying process. Regarding the SEM-based mediation analysis, we could certainly have used it to compare the proportion of mediated effect obtained through SEM and the % of change obtained through this method. We decided to use the percentage of reduction in de RR in order to make our results more comparable with other studies in the field. However, we are now aware of the increasing use of SEM in the field and therefore will definitely consider this approach in future projects.

---

## [Decision Letter · Decision Letter 1]

20 Mar 2023

Maternal educational level and preterm birth: Exploring inequalities in a hospital-based cohort study

PONE-D-22-22954R1

Dear Dr. Granés,

We’re pleased to inform you that your manuscript has been judged scientifically suitable for publication and will be formally accepted for publication once it meets all outstanding technical requirements.

Kind regards,

Simone Garzon

Academic Editor

PLOS ONE

Additional Editor Comments (optional):

Reviewers' comments:

Reviewer's Responses to Questions

**Comments to the Author**

1. If the authors have adequately addressed your comments raised in a previous round of review and you feel that this manuscript is now acceptable for publication, you may indicate that here to bypass the “Comments to the Author” section, enter your conflict of interest statement in the “Confidential to Editor” section, and submit your "Accept" recommendation.

Reviewer #1: All comments have been addressed

Reviewer #2: All comments have been addressed

2. Is the manuscript technically sound, and do the data support the conclusions?

Reviewer #1: Yes

Reviewer #2: Yes

3. Has the statistical analysis been performed appropriately and rigorously? 

Reviewer #1: Yes

Reviewer #2: Yes

4. Have the authors made all data underlying the findings in their manuscript fully available?

Reviewer #1: Yes

Reviewer #2: (No Response)

5. Is the manuscript presented in an intelligible fashion and written in standard English?

Reviewer #1: Yes

Reviewer #2: Yes

6. Review Comments to the Author

Reviewer #1: Most issues were resolved with the first round of revision and the article is now acceptable for publication.

Reviewer #2: (No Response)

7. PLOS authors have the option to publish the peer review history of their article (what does this mean?). If published, this will include your full peer review and any attached files.

Reviewer #1: No

Reviewer #2: **Yes: **Susanne Hesselman

---

## [Editor Report · Acceptance letter]

27 Mar 2023

PONE-D-22-22954R1 

Maternal educational level and preterm birth: Exploring inequalities in a hospital-based cohort study 

Dear Dr. Granés:

I'm pleased to inform you that your manuscript has been deemed suitable for publication in PLOS ONE. Congratulations! Your manuscript is now with our production department. 

Kind regards, 

on behalf of

Dr. Simone Garzon 

Academic Editor

PLOS ONE